# Consultation-Based Deprescribing Service to Optimize Palliative Care for Terminal Cancer Patients

**DOI:** 10.3390/jcm12237431

**Published:** 2023-11-30

**Authors:** Minoh Ko, Sunghwan Kim, Sung Yun Suh, Yoon Sook Cho, In-Wha Kim, Shin Hye Yoo, Ju-Yeun Lee, Jung Mi Oh

**Affiliations:** 1College of Pharmacy and Research Institute of Pharmaceutical Sciences, Seoul National University, Seoul, Republic of Korea; minovill@snu.ac.kr (M.K.); iwkim@snu.ac.kr (I.-W.K.); jypharm@snu.ac.kr (J.-Y.L.); 2Department of Pharmacy, Seoul National University Hospital, Seoul, Republic of Korea; 30349@snuh.org (S.K.); 30208@snuh.org (S.Y.S.); joys99@snuh.org (Y.S.C.); 3Center for Palliative Care and Clinical Ethics, Seoul National University Hospital, Seoul, Republic of Korea; ifi1024@snu.ac.kr

**Keywords:** deprescribing, hospitalized patients, consultation-based palliative care, end-of-life care, pharmaceutical care

## Abstract

(1) Background: A pharmacist-led deprescribing service previously developed within the Consultation-Based Palliative Care Team (CB-PCT) was implemented for terminal cancer patients. (2) Objective: To evaluate the clinical outcomes of the developed deprescribing service for terminal cancer patients in CB-PCT. (3) Methods: A retrospective analysis compared the active care (AC) group to the historical usual care (UC) group. The clinical outcomes included the deprescribing rate of preventive medications, the proportion of patients with one or more medication-related problems (MRPs) resolved upon discharge, and the clinical significance. The implementability of the service was also gauged by the acceptance rates of pharmacists’ interventions. (4) Results: Preventive medications included lipid-lowering agents, gastroprotective agents, vitamins, antihypertensives, and antidiabetic agents. The AC group revealed a higher deprescribing rate (10.4% in the UC group vs. 29.6% in the AC group, *p* < 0.001). At discharge, more AC patients had one or more MRPs deprescribed (39.7% vs. 2.97% in UC, *p* < 0.001). The clinical significance consistently had a very significant rating (mean score of 2.96 out of 4). Acceptance rates were notably higher in the AC group (30.0% vs. 78.0%. *p* = 0.003). (5) Conclusions: The collaborative deprescribing service in CB-PCT effectively identified and deprescribed MRPs that are clinically significant and implementable in practice.

## 1. Introduction

Terminally ill cancer patients, who have a life expectancy of less than six months, often contend with numerous comorbidities and systemic symptoms. The pivotal challenges lie in effectively managing these symptoms while maintaining their quality of life (QoL). However, the administration of numerous prescriptions to alleviate multifarious symptoms may inadvertently compromise the patients’ QoL. Notably, in Korea, even with the introduction of the Hospice and Palliative Care and Self-determination Life Sustaining Treatment Act in 2018, the utilization of hospice care remains limited to only 20% of this terminally ill cancer patients [1,2].

In 2015, recognizing that many patients are first informed of their terminal status while hospitalized due to acute events and subsequently recommended for hospice care, the Korean National Health Insurance Service (NHIS) introduced reimbursement for consultation-based palliative care team (CB-PCT) services [1]. This initiative was designed to ensure patients are transitioned timely to the most appropriate care settings and to enhance the overall quality of care, with an emphasis on palliative needs. CB-PCT is a multidisciplinary team of palliative care experts, including doctors, nurses, social workers, and pharmacists. They provide comprehensive palliative care to patients in general or acute care wards. The team delivers specialized care to terminal patients who either do not wish to be admitted to a palliative care ward or are waiting placement. Their role is particularly valuable in hospitals without dedicated hospice wards, serving as a bridge to facilitate access to hospice care. However, at its inception, pharmacists were not considered as mandatory service providers. Nonetheless, providing comprehensive pharmaceutical care to these patients is crucial for effectively managing polypharmacy (PP), addressing suboptimal symptoms, optimizing medication utilization, and preventing legacy prescription from prescribing cascades. Terminal patients encounter various pharmacokinetic challenges, such as alterations in the volume of distribution due to reduced serum albumin levels. Additionally, reductions in cytochrome 2E1 and 2D6 have been observed, affecting the clearance of several drugs [3]. PP, a significant concern resulting from aggressive treatments in geriatric and terminally ill cancer patients, is a predominant pharmaceutical care challenge. This is due to its association with an increased pill burden, especially from preventive medications [4,5,6]. Previous studies have indicated that approximately 45% of this demographic experiences PP, a figure that escalates to over 60% for those in the terminal stages of cancer [7,8]. The correlation between PP and an elevated risk of adverse drug events, usage of potentially inappropriate medications (PIMs), hospitalizations, and emergency room visits is well-established [9,10,11,12]. Moreover, studies underline that higher instances of PP are correlated with an intensified burden of symptoms and diminished QoL [13].

Deprescribing has emerged as a promising strategy for managing PP [14]. This patient-centered approach, which involves the discontinuation or de-escalation of unnecessary medications, seeks to address PP and improve patient outcomes [15,16,17,18]. The significance of deprescribing is also acknowledged not only by healthcare professionals but also by patients and caregivers. One study revealed that over 70% of patients wished to reduce the number of prescriptions, if feasible [19]. Consequently, guidelines such as OncPal [20] and STOPPFrail [21] have been developed to facilitate improved outcomes for advanced cancer patients. Though these guidelines are comparably effective, noble distinction exists between them [22], suggesting that a holistic approach—melding multiple guidelines—could yield optimal results in practical applications [23,24]. Several pharmacist-led deprescribing service models have demonstrated their efficacy in reducing PIMs and medication-related problems (MRPs) while adeptly managing symptoms [25,26,27,28,29]. To deliver efficient and effective medication services to patients, it is imperative to understand the types, causes, and causative medications of medication-related problems (MRPs) that the target patients possess. For precise classification, the Pharmaceutical Care Network Europe (PCNE) has established a system for categorizing drug-related problems. The PCNE classification has been updated to version 9.1 and is composed of the problem, cause, intervention, and acceptance of the intervention. Within the intervention category, it further subdivides into detailed levels such as prescriber level, patient level, and drug level. Network analysis, a method that discerns relationships and patterns among various nodes within a network, can be effectively utilized for elucidating the relationship more clearly [30].

Building upon these insights, in 2019, we developed deprescribing guidelines and established a pharmacist-led deprescribing service in CB-PCT. The blueprint for the pharmacist-led deprescribing service within the consultation-based palliative care team was based on the “4D” framework, Discover, Define, Design, and Develop, a methodology previously employed by Han et al. in the DrugTEAM^TM^ study group [31]. 

The primary aim of this study is to implement and evaluate the clinical outcomes of a developed multidisciplinary deprescribing service in the CB-PCT setting.

## 2. Materials and Methods

### 2.1. Pharmacist-Led Deprescribing Service of CB-PCT 

A previously designed collaborative deprescribing program for terminally ill cancer patients was put in service as consultation-based palliative care in two phases at Seoul National University Hospital (SNUH). The team was comprised of two physicians, one nurse, one social worker, and one pharmacist, with the pharmacist having over three years of experience in oncology pharmaceutical care. The target populations for this pharmaceutical service included patients enrolled in the consultation-based palliative care team, along with healthcare providers from the wards caring for these patients, and the members of the palliative care team itself. Furthermore, there were four objectives for this deprescribing pharmaceutical service. First, it aimed to provide medication reconciliation to accurately ascertain the patient’s medication history and minimize unintentional discrepancies during transition of care. Second, by offering comprehensive medication evaluation and deprescribing services, the goal was to identify and resolve medication-related problems in terminally ill cancer patients and to deprescribe medications with a low risk-benefit ratio. Third, it involved providing evidence-based drug information services to deliver accurate medical information to healthcare professionals. Finally, its objective was to ensure the continuity of pharmacist interventions through the application of discharge pharmaceutical care services.

In the initial phase, spanning from October 2019 to February 2020, patients were provided with usual care (UC). For those in the UC group, pharmacists evaluated the medications and conducted one-time deprescribing intervention during the CB-PCT weekly meetings. In this phase, pharmacists reviewed patients’ medication profiles only upon their enrollment in the CB-PCT service, and interventions were exclusively made at the prescriber level. After the initial phase, from March 2020 to July 2020, patients received active care (AC), which included the pharmacist-led deprescribing service within the CB-PCT (for an elucidation regarding service development, refer to Appendix A). The AC service was delivered by the same pharmacists who delivered the UC group. Pharmacists communicated with the CB-PCT team members through an instant messaging platform. In addition to participating in the weekly CB-PCT meetings, pharmacists also provided the following sub-services (Figure 1): Medication Reconciliation (MR) service: This sub-service aimed to curtail medication discrepancies and obtain the best possible medication history (BPMH) within 48 h of patients’ admission or enrollment in CB-PCT.Comprehensive Medication Evaluation and Deprescribing (CME&D): Initiated within 48 h post CB-PCT enrollment, this core sub-service continued throughout the patients’ stay. CME&D offered deprescribing interventions and an in-depth medication assessment to identify MRPs, which include both evident problems and potentially inappropriate prescriptions. CME&D involved stages of finding, assessing, discussing, recommending, monitoring, and documenting. During the finding stage, patients’ medications were evaluated and pharmacy consultations were provided to patients as needed, with pharmacists typically spending about 15 min with each patient. In the assessing stage, opportunities for deprescribing were identified primarily based on the previously developed deprescribing guidelines (SNUH deprescribing guidelines, Appendix A) and a variety of established guidelines, including Beers’ criteria [32] and STOPPFrail [21]. Detected MRPs were shared within the team and actively discussed, not only during CB-PCT weekly meetings, but also via the team’s instant messaging apps, ensuring timely interventions. Pharmacists intervened via short message service to physicians or through verbal communication. Following evaluation, the findings were documented in the electronic health records (EHR) using department-specific forms. Pharmacists evaluated patients’ prescriptions within 24 h of notification by the team’s nurse of the patient’s enrollment. Aligned with the patient’s consultation cycle of the team, MRPs were tracked and monitored for resolution. Finally, the results of the interventions were recorded in the electronic health record and pharmacist’s database.Evidence-based Drug Information (EB-DI): A service proffering optimal drug information to healthcare providers.Discharge Pharmaceutical Care Transition (DPCT): Implemented at discharge, the primary objective of DPCT was to facilitate the transition by addressing the pharmaceutical care needs tailored for these patients. Pharmacists reviewed the appropriateness of prescribed medications, focusing on optimizing symptom management for the patient. The continued necessity of previously used medications was reassessed, and recommendations for medication adjustments and deprescribing were given for the prescribed discharge medications.

During the period of the AC group intervention, the COVID-19 pandemic began to emerge. At that time, South Korea managed to keep the number of confirmed COVID-19 cases relatively low compared to other countries. However, in accordance with national quarantine guidelines, the weekly CB-PCT meetings were adjusted to be held every two weeks.

### 2.2. Study Design

A retrospective cohort study was conducted, comparing the AC group with the historically controlled UC group. Eligible participants were terminally ill patients aged 20 or older who were diagnosed with advanced solid or hematologic malignancies and enrolled in the CB-PCT at SNUH between 1 October 2019 and 31 July 2020. Individuals who withdrew, died, were transferred, or discharged prior to receiving the CB-PCT service were excluded. This study was approved by the Institutional Review Board of the hospital (SNUH IRB No. 2103-031-1201).

Demographic characteristics, oncologic details, and medication specifics of patients were sourced from the electronic health records. Upon enrollment in the CB-PCT, we collected pertinent demographic information including patient’s age, sex, comorbid diseases, performance status based on Eastern Cooperative Oncology Group (ECOG) score, and nutrition intake methods. Common comorbidities primarily included common geriatric illnesses, such as hypertension, diabetes mellitus, dyslipidemia, ischemic heart disease, and chronic obstructive pulmonary diseases. Oncologic information included diagnosis, chemotherapy history, and the continuation of chemotherapy post-CB-PCT enrollment. Regarding medication data, we compiled medication profiles both before and after the pharmacist’s interventions. Using comprehensive prescription lists, we calculated the number of prescriptions for each drug class, the number of oral medications, and the total number of prescriptions. In the AC group, pharmacists consulted with the patients, allowing them to identify over-the-counter (OTC) medications that the patients were taking. However, in the UC group, since there were no consultations, information about OTC medications could not be obtained. Medication-related problems (MRPs) and corresponding pharmacist’s interventions (PIs) were extracted from the pharmacist record database. Both MRPs and PIs were evaluated and categorized in accordance with the Pharmaceutical Care Network Europe (PCNE) classification version 9.1 [33]. PP was defined as the use of five or more medications. Deprescribing interventions were defined as either discontinuation or tapering off the unnecessary medications. 

This study evaluated clinical outcomes and implementability of the service.

#### 2.2.1. Clinical Outcomes

To assess the clinical outcomes of the service, the deprescribing rate of preventive medications was calculated and compared between the UC and AC groups. Medications that were frequently used and identified as preventive based on indication analysis were selected as preventive medications. Moreover, drug classes deemed futile in a study of Korean terminal cancer patients were also categorized as preventive medications [34]. The deprescribing rate of preventive medications was calculated based only on those prescriptions identified for preventive use following the pharmacist’s evaluation of their indications.

In addition, the proportion of patients resolved one or more MRPs upon discharge, the changes in pill burden, and the changes in the proportion of patients experiencing PP at the time of discharge were evaluated and compared between the groups. The deprescribing rate was determined by dividing the number of prescriptions deprescribed post-intervention by the total number of prescribed preventive medications. Change in pill burden referred to the shifts in the median number of oral medications taken before and after the deprescribing service.

To evaluate the clinical significance of the service, we complied a consolidated list of cases intervened to resolve MRPs. The value of the pharmacist’s deprescribing service was reviewed and rated by a panel of professionals using a six-point Likert scale, adapted from the criteria by Overhage et al. [35] (Table 1). The professional panel comprised of physicians specializing in palliative care and oncology pharmacists with over five years of experience.

#### 2.2.2. Implementability

To assess the implementability of the service, acceptance rates (%) of PIs were assessed at the prescriber, patient, and drug levels. PIs were considered as accepted when the MRPs were resolved.

### 2.3. Statistical Analysis and Network Analysis

Descriptive statistics were utilized to depict the categories of MRPs, PIs, acceptance rate (%), and the clinical significance of PIs. In comparing the UC and AC groups, a Wilcoxon rank-sum test was used for continuous variables, as determined by the Shapiro–Wilk test outcomes. For categorical variables, the Chi-square or Fisher’s exact test was employed. To assess the inter-rater reliability of the survey responses, Kendall’s W, a non-parametric statistic for rank correlation, was computed [36]. All statistical analyses were performed using R-software (R for Windows 4.2; The R Foundation for Statistical Computing, Vienna, Austria) and R-studio version 3.0, with a significance level of 0.05. 

A network analysis was undertaken to identify the most prevalent MRP and explore the association between causative drugs, MRP causes, and problems. To visually represent this network, we employed the Harel–Koren Fast Multiscale layout algorithm using Network Overview, Discovery, and Exploration for Excel (NodeXL), a Microsoft Excel software package. The resulting graph was presented in a directed format, effectively capturing the relationships between the vertices. To identify the primary causative agents and causes, we calculated a range of centrality metrics, which included in-degree centrality (CD-in), out-degree centrality (CD-out), betweenness centrality, and closeness centrality. These metrics determined the size of the vertices and the thickness of the edges in the visual representation. Additionally, through visual inspections of the sociogram we sought further insights into these associations. Separate sociograms were generated for interventions that were accepted and those that were rejected, providing a comprehensive view of the network dynamics.

## 3. Results

### 3.1. Patient Characteristics

A total of 275 patients participated in this study, comprising 101 in the UC group and 174 in the AC group. Five patients (one from the UC group and four from the AC group) expired within 48 h of enrollment and were consequently excluded from the analysis. Overall, the two groups had comparable baseline characteristics, including proportion of patients with performance status (PS) scores of 3 or more (97.0% in UC vs. 93.5% in AC, *p* = 0.61) (Table 2). The median durations from the day of enrollment in the CB-PCT to the day of discharge were 7 days (UC) and 9 days (AC). The most prevalent cancer types were non-small cell lung cancer and pancreatic cancer. Additionally, similar proportions of end-of-life patients were included, accounting for 42% in the UC group and 41.8% in the AC group. 

Medication profiles of terminal cancer patients enrolled in CB-PCT were assessed. At baseline assessment, pill burden was slightly higher in the AC group (median [minimum–maximum], 4 [1–7] for UC compared to 5 [2–8] in AC, *p* = 0.046). The prevalence of PP was approximately 90% in both groups (90.0% in the UC vs. 90.6% in the AC, *p* = 1.00). In both groups, at least 40% of patients were administering 10 or more medications (39.0% vs. 50.0%, *p* = 0.10) (Table 3). Over 80% of patients were prescribed pain managing medications. Additionally, 40% of patients were on gastroprotective agents and 52.7% were taking vitamins. Both gastroprotective agents and vitamins are significant preventive medications that can potentially contribute to the pill burden for palliative cancer patients. One fifth of the patients were taking liver protective agents, which are among the major drug classes that patients often take for extended periods without recent re-evaluation of their appropriateness or necessity. The proportion of patients taking antidiabetic medications closely matched the proportion of those diagnosed with diabetes. (Figure 2).

### 3.2. Clinical Outcomes

Various MRPs were identified and addressed during the deprescribing service. A total of 2606 prescriptions were examined, encompassing 907 from the UC group and 1699 from the AC group. A significantly higher proportion (%) of patients in the AC group (104 (61.2%)) had at least one MRP identified compared to the UC group (37 (37.0%), *p* < 0.001). Correspondingly, a greater number of MRPs were identified in the AC group (1.15 MRPs per patient) in comparison to the UC group (0.16 MRPs per patient). The details on identified MRPs are presented in Appendix A. In both groups, the primary contributors to MRPs were suboptimal use of pain managing medications, such as low doses of analgesics and omission of opioids, when indicated. This was followed by preventive use of lipid-lowering agents and gastroprotective drugs (Figure 3). The results of network analysis and sociograms detailing significant causative drugs and types of MRPs are presented in Table 4 and Figure 4. The leading causes of MRPs that exhibited significant in-degree centrality (CD-in), a metric measuring the extent of connections directed towards specific nodes within a network, were identified. These causes included instances where there was no proper indication for drug usage (C1.2) characterized by CD-in of 15. (Appendix A). Lipid lowering agents were the major preventive medications that contributed to the pill burden. Gastroprotective agents, vitamins, antihypertensives, and antidiabetic agents were selected as they were the top five preventive medications prescribed at baseline. 

A notably higher proportion of patients in the AC group experienced positive outcomes, with preventive medications being deprescribed in 29.6% of patients, compared to 10.4% in the UC group (*p* < 0.001) (Table 5). Additionally, 40.0% in the AC group had one or more MRPs deprescribed at discharge, in contrast to only 3.00% in the UC group (*p* < 0.001). The change of pill burden, calculated by comparing the number of oral medications before and after deprescribing interventions was not significantly different (median [minimum-maximum], 0 [−9, 10] in the UC vs. 0 [−11, 7] in AC group, *p* = 0.13); similarly, the change in proportion of patients with PP at discharge showed no significant difference (−14% in the UC group vs. −8.1% in the AC group, *p* = 0.87) (Table 6).

Overall, the clinical significance of the deprescribing service was evaluated as very significant, with a mean score of 2.96 (standard deviation (SD), 0.80) on a scale ranging from −1 to 4. Importantly, physicians valued the service significantly more than the pharmacists, giving it a score of 3.22 [SD, 0.31] (very significant~extremely significant), whereas pharmacists rated it with a score of 2.70 [SD, 0.33] (significant~very significant) out of 4. The agreement levels between physicians and pharmacist were measured using Kendall’s W statistics, revealing fair agreement among physicians (Kendall’s W 0.35) and moderate agreement among pharmacists (Kendall’s W 0.47) [36].

### 3.3. Implementability

In the AC group, there was a significant increase not only in the number of PIs but also the acceptance rates (%) at all levels. Specifically, at the prescriber level, the acceptance rate was higher in the AC group (68.8% in UC vs. 78.9% in AC, *p* = 0.021). Moreover, there was an increase in proposals to the prescriber (I1.3) during the active deprescribing period (55.6% in UC vs. 81.8% in AC, *p* = 0.07) (Table 7, detailed frequency and acceptance rate (%) for each MRP in Appendix A). Similarly, at the drug level, the acceptance rate was significantly higher in the AC group (30.0% in UC vs. 78% in AC, *p* = 0.003). Within the AC group, high acceptance rates were observed across various drug classes, including pain management agents (92.5%), vitamins and minerals (91.7%), and antihypertensives (90%, Appendix A). In contrast, the acceptance rates were lowest for deprescribing involving chemotherapeutic agents and anti-infectives for active symptom controls (33.3% and 54.6%, respectively).

## 4. Discussion

In this retrospective study, we implemented a collaborative deprescribing service within the CB-PCT framework. The primary focus was to evaluate clinical effectiveness and significance of this service for terminally ill cancer patients in CB-PCT. This study suggests that the deprescribing service within CB-PCT can effectively reduce preventive medicines and minimize MRPs. The findings of this study revealed a high prevalence of PP in terminally ill cancer patients, with over 90% of patients prescribed five or more medications. Intriguingly, nearly 53% of all patients were taking gastroprotective agents. Additionally, the preventive medications addressed in this study, including gastroprotective agents, were also identified as futile medications in a previous study [34]. Specifically, 80% of the MRPs associated with gastroprotective agents were due to either lack of indication or unnecessary drug treatment. The insights gathered from our study have the potential to inform medication utilization for terminally ill cancer patients, contributing to further research endeavors and the development of public health policies related to end-of-life care. It is worth noting that the number of medications at baseline was slightly higher in the AC group. This difference may be attributed to pharmacists obtaining lists of OTC medications and herbal remedies in the AC group. 

A pharmacist-led multidisciplinary approach for deprescribing decisively identified MRPs. This suggests that active involvement of pharmacists in patient care plays a pivotal role in resolving MRPs. While the median change in pill burden did not significantly differ between the groups, it is noteworthy that some patients were able to discontinue more than ten medications, effectively reducing their pill burden. Any observed increases in medication use were primarily intended for symptom management. The most common cases involved the addition of appropriate analgesics to address suboptimal pain management in patients. While reducing a patient’s pill burden is crucial, even more imperative is optimizing medication use by minimizing unnecessary drugs and focusing on those that manage patient’s symptoms. Hence, we assessed the clinical significance of PIs through the expertise of palliative care, which was considered very significant. Previous studies [37,38] assessed the clinical significance of pharmaceutical care services with input from a single representative of each healthcare profession. However, to enhance the reliability, our study involved evaluations from five palliative care specialist physicians and five clinical pharmacists to ascertain the value of the service. An interesting observation from our study was the slightly differing perspectives of physicians and pharmacists regarding the clinical significance of PIs. Physicians rated PIs as ‘very significant’ to ‘extremely significant’, while pharmacists rated them as ‘significant’ to ‘very significant’. This suggests that pharmacists underestimate the importance of their role in palliative patient care. 

Our significant outcome of the active deprescribing service was the increased proportion of the AC group patients in whom one or more MRPs were resolved upon discharge. This suggests that pharmacist-led interventions focused on deprescribing, guided by established guidelines, exhibited notable effectiveness. Interestingly, the AC group displayed a marginally higher proportion of patients undergoing cancer treatment and discharged to their homes, indicating a greater tendency toward persistent use preventive medications, and explaining the higher identification of MRPs in this group.

The significantly higher acceptance rate was observed in the AC group, which underscores significant improvement in implementability within the AC group across all levels. The interventions were accepted especially at the prescriber and drug levels. This indicates a greater receptivity and alignment with the proposed interventions. The drug class with the highest acceptance rate was pain managing medications, which indicates the importance of optimal management of the symptoms. High acceptance rate of noteworthy preventive medications such as vitamins and antihypertensives suggests the implementability of deprescribing service. 

Our study contributes to the growing field of deprescribing by shedding light on the need for deprescribing in terminally ill cancer patients. A previous study [39] conducted in a single center demonstrated the successful reduction of prescriptions through pharmacist-led, collaborative deprescribing service, but did not explicitly evaluate the MRPs or clinical significance of the PIs. In comparing MRPs between two groups, we found a strong need for deprescribing in terminally ill cancer patients. The main causative agents, which were lipid lowering agents and opioids, identified in this study were also found to be the most common types of inappropriate contributors to PP in this population [4,40]. This suggests that, for end-of-life cancer patients, emphasis should be on minimizing the use of preventive medications and prioritizing the management of significant symptoms such as pain [41]. Gastroenterologists often prefer to continue prescribing prokinetics, even when patients lack compelling indications, such as intestinal obstructions. 

The consultation-based palliative care model is recognized for its effectiveness in delivering palliative care with limited resources. This approach has been a foundational aspect of many early palliative care models in the United States. At our institution, nutritional management serves as another exemplary implementation of this consultative model [42]. This indicates the versatility of the consultation-based model in addressing diverse patient care needs, demonstrating its broad applicability across different aspects of patient care. Moreover, for future implementation of the service, factors associated with MRPs were identified (the results presented in the Appendix A). These factors include comorbidities such as diabetes, ischemic heart disease, ongoing anti-cancer treatment, and a high pill burden. Additionally, immediate referral to hospice care after a cancer diagnosis also emerged as a risk factor. Notably, rapid cancer progression was found to contribute to higher MRPs, which is a novel finding compared to previous studies [7,43]. 

There are some limitations for this study. Like all retrospective studies, potential bias needs to be considered. Moreover, challenges in monitoring long-term outcomes may arise due to instances of patients’ deaths or patients being lost to follow-up. The reluctance of some physicians to deprescribe medications for these patients highlights the need to promote broader acceptance of deprescribing concepts in Korea through further studies. 

This study is the first to implement a pharmacist-led collaborative deprescribing service model for terminally ill patients within CB-PCT in Korea. This study not only underlines the urgent need for deprescribing medications that are no longer beneficial or may even be causing harm to terminally ill cancer patients, but also emphasizes the crucial role that pharmacists can play in improving symptom managements.

## 5. Conclusions

This study evaluated the clinical outcomes of a pharmacist-led deprescribing service for terminally ill cancer patients within CB-PCT settings in Korea. The AC group demonstrated a notably higher deprescribing rate for preventive medication and a significant increase in the proportion of patients with one or more MRPs being resolved upon discharge. The clinical significance of the service was rated as very significant with a noteworthy improvement of implementability.

## Figures and Tables

**Figure 1 jcm-12-07431-f001:**
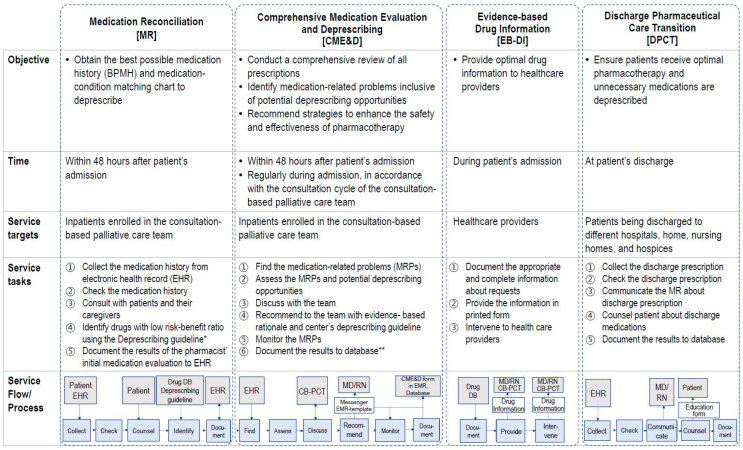
Active care model and flow for deprescribing service in consultation-based palliative care team (CB-PCT). * Center’s deprescribing guidelines indicates the developed deprescribing guidelines in the study. ** Database: database shared between pharmacists. BPMH, best possible medication history; CB-PCT, consultation-based palliative care team; CME&D, comprehensive medication evaluation and deprescribing; DB, database shared between pharmacists; DPCT, discharge pharmaceutical care transition; EB-DI, evidence-based drug information; EHR, electronic medical record; MD, medical doctors; MRP, medication-related problems; RN, registered nurse.

**Figure 2 jcm-12-07431-f002:**
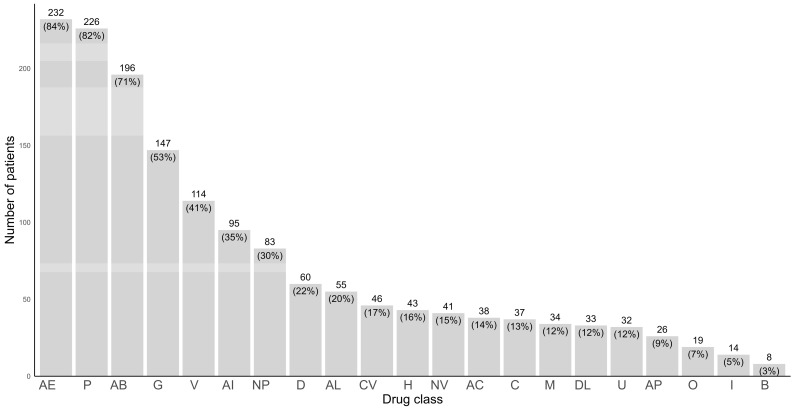
Drug class utilization among terminally ill cancer patients upon enrollment of the CB-PCT. AB, anti-infectives; AC, anticoagulants; AE, complementary therapies; AI, hormones; AL, liver supplements; AP, antiplatelet agents; B, parasympathomimetics; C, chemotherapeutic agents; CV, medications to treat cardiovascular diseases (except for antihypertensives); D, hypoglycemic agents; DL, lipid lowering agents; G, gastroprotective agents; H, antihypertensives; I, immunosuppressants; M, prokinetics; NP, neuropsychiatric drugs; NV, antiemetics; O, osteoporosis medications; P, pain managing agents; U, drugs for urinary frequency and incontinence; V, vitamins and minerals.

**Figure 3 jcm-12-07431-f003:**
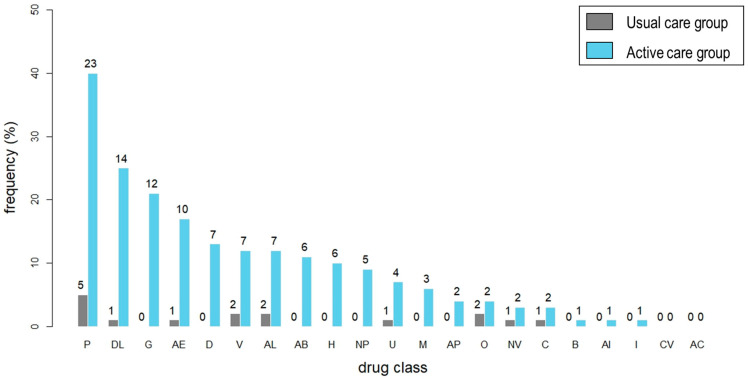
Proportion (%) of medication-related problems (MRPs) identified by drug class in each group. AB, anti-infectives; AC, anticoagulants; AE, complementary therapies; AI, hormones; AL, liver supplements; AP, antiplatelet agents; B, parasympathomimetics; C, chemotherapeutic agents; CV, medications to treat cardiovascular diseases (except for antihypertensives); D, hypoglycemic agents; DL, lipid lowering agents; G, gastroprotective agents; H, antihypertensives; I, immunosuppressants; M, prokinetics; NP, neuropsychiatric drugs; NV, antiemetics; O, osteoporosis medications; P, pain managing agents; U, drugs for urinary frequency and incontinence; V, vitamins and minerals.

**Figure 4 jcm-12-07431-f004:**
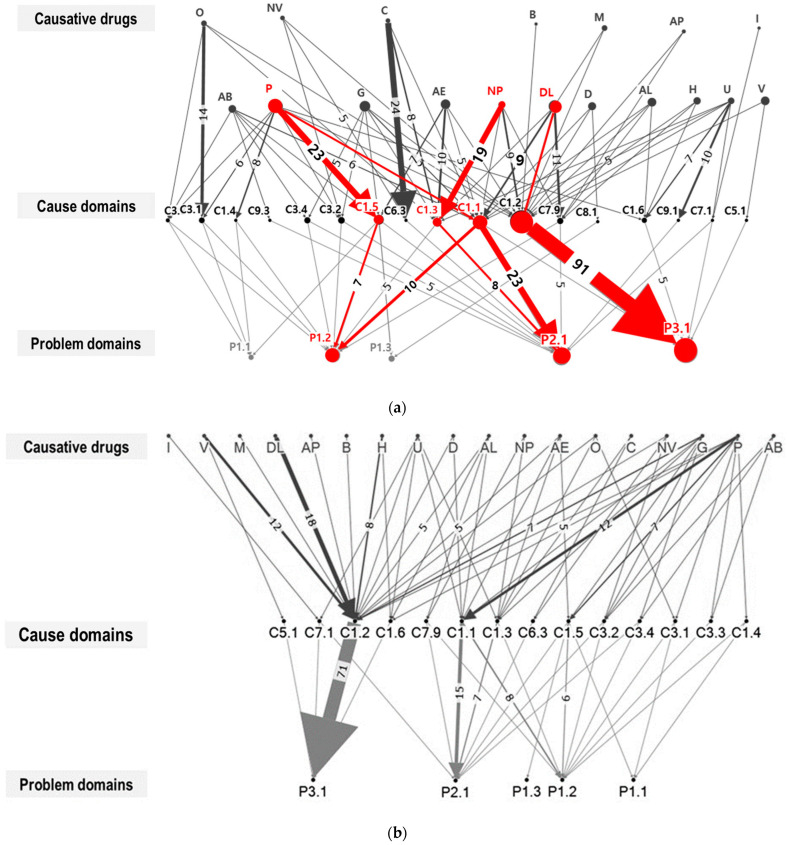
(**a**) Sociogram of causative drugs, cause domains, and problem domains of total MRPs; (**b**) Sociogram of causative drugs, cause domains, and problem domains of MRPs accepted after pharmacists’ interventions; (**c**) Sociogram of causative drugs, cause domains, and problem domains of MRPs NOT accepted after pharmacist’s interventions. • Size of the vertices and width of edges represent the incidence of each item, and only edges with counts of 5 or more were labelled. • Causative drugs: AB, Anti-infectives; AC, Anticoagulants; AE, Complementary therapies; AI, Hormones; AL, Liver supplements; AP, Antiplatelet agents; B, Parasympathomimetics; C, Chemotherapeutic agents; CV, Medications to treat cardiovascular diseases (except for antihypertensives); D, Hypoglycemic agents; DL, Lipid lowering agents; G, Gastroprotective agents; H, Antihypertensives; I, Immunosuppressants; M, Prokinetics; NP, Neuropsychiatric drugs; NV, Antiemetics; O, Osteoporosis medications; P, Pain managing agents; U, Drugs for urinary frequency and incontinence; V, Vitamins and minerals. • Causative domains: C1.1, Inappropriate drug according to guidelines/formulary; C1.2, No indication for drug; C1.3, Inappropriate combination of drugs, or drugs and herbal medications, or drugs and dietary supplements; C1.4, Inappropriate duplication of therapeutic group or active ingredient; C1.5, No or incomplete drug treatment in spite of existing indication; C1.6, Too many different drugs/active ingredients prescribed for indication; C3.1, Drug dose too low; C3.2, Drug dose of a single active ingredient too high; C3.3, Dosage regimen not frequent enough; C3.4, Dosage regimen too frequent; C5.1, Prescribed drug not available; C6.3, Drug over-administered by a health professional; C7.1, Patient intentionally uses/takes less drug than prescribed or does not take the drug at all for whatever reason; C7.9, Patient physically unable to use drug/form as directed; C8.1, Medication reconciliation problem; C9.1, No or inappropriate outcome monitoring; C9.3, No obvious cause. • Problem domains: P1.1, No effect of drug treatment despite correct use; P1.2, Effect of drug treatment not optimal; P1.3, Untreated symptoms, or indication; P2.1, Adverse drug event (possibly) occurring; P3.1, Unnecessary drug-treatment; P3.2, Unclear problem/complaint.

**Table 1 jcm-12-07431-t001:** Criteria used for assessing clinical significance (adapted from the criteria by Overhage et al.).

Value of Service	Score	Criteria
Extremely significant	4	Recommendation qualified by extremely serious consequences or potential life-and-death situation
Very significant	3	Recommendation qualified by a potential or existing dysfunction in a major organ or avoidance of serious adverse drug interaction or contraindication to use
Significant	2	Recommendation would bring patient care to a more acceptable, appropriate level (i.e., standard of practice), including quality-of-life issues with evidence from the patient or documentation elsewhere, as well as issues of cost and convenience.
Somewhat significant	1	Patient’s benefit from the recommendation could be neutral depending on professional interpretation or more information or a clarification must be obtained by the pharmacist from the physician, nurse, or other appropriate health care professional before an order can be processed
No significance	0	Information only or recommendation not patient specific
Adverse significance	−1	Recommendation inappropriate; its implementation may lead to adverse outcomes

**Table 2 jcm-12-07431-t002:** Baseline characteristics of palliative care patients enrolled in consultation-based palliative care team (CB-PCT).

Characteristics	UC Group (n = 100)	AC Group (n = 170)	*p*-Value
Age (yr) ^a^	66.5 [57–74.5]	63 [56–72]	0.22
Sex (Female)	47 (47.0%)	75 (44.1%)	0.74
Cancer diagnosis			0.12
Non-small cell lung cancer	13 (13.0%)	37 (21.8%)	
Pancreatic cancer	17 (17.0%)	19 (11.2%)	
Others *	70 (70.0%)	114 (67.1%)	
Co-morbid diseases			
Hypertension	28 (28.0%)	43 (25.3%)	0.73
Diabetes	19 (19.0%)	44 (25.9%)	0.25
Dyslipidemia	10 (10.0%)	13 (7.6%)	0.66
Ischemic heart disease	30 (30.0%)	51 (30.0%)	1.00
Chronic obstructive pulmonary disease	4 (4.0%)	8 (4.7%)	1.00
Performance status (ECOG score)			
≥3	97 (97.0%)	159 (93.5%)	0.61
Diet			
NPO	15 (15.0%)	21 (12.4%)	0.28
On total parenteral nutrition	65 (65.0%)	97 (57.1%)	0.18
History of chemotherapy	91 (91.0%)	160 (94.1%)	0.45
Continuation of cancer treatment			0.06
Chemotherapy	16 (16.0%)	46 (27.1%)	
Radiotherapy	4 (4.0%)	3 (1.8%)	
Follow-up duration ^a※^	7.0 [4.0–14.0]	9.0 [4.0–14.0]	0.41
Type of discharge			*0.02*
Expiration	42 (42.0%)	71 (41.8%)	
Places other than home ^†^	42 (42.0%)	49 (28.8%)	
Home	16 (16.0%)	50 (29.4%)	
Department requesting consultation			0.30
Hemato-oncology	75 (75.0%)	116 (68.2%)	
Non hemato-oncology	25 (25.0%)	54 (31.8%)	

^a^ median [minimum–maximum]. AC, active care; ECOG, Eastern Cooperative Oncology Group; NPO, nil per oral; UC, usual care; ^※^ Durations from the day of enrollment in the CB-PCT to the day of discharge. * Others include advanced gastric cancer, angioimmunoblastic T cell lymphoma, acute myeloblastic leukemia, Burkitt lymphoma, bone sarcoma, brain tumor, cholangiocarcinoma, colorectal cancer, cervical cancer, diffuse large B cell lymphoma, NK-T cell lymphoma, primary CNS lymphoma, renal cell carcinoma, small cell lung cancer, soft tissue sarcoma, thyroid cancer, urothelial cancer, endometrial cancer, esophageal cancer, gall bladder cancer, hepatocellular carcinoma, Hodgkin’s lymphoma, head and neck cancer, metastatic breast cancer, melanoma, multiple myeloma, malignant pleural mesothelioma, and multiple primary tumors. ^†^ Places other than home include local hospital, nursing home, and hospice.

**Table 3 jcm-12-07431-t003:** Medication profiles of palliative care patients enrolled in consultation-based palliative care team (CB-PCT).

Characteristics	UC Group (n = 100)	AC Group (n = 170)	*p*-Value
**Number of medications**			
Number of total medications in use at the time of enrollment ^a^	9 [6–11]	9.5 [7–13]	0.04
Number of oral medications in use at the time of enrollment ^a^	4 [1–7]	5 [2–8]	0.05
**Proportion (%) of polypharmacy**			
Proportion (%) of patients administering 5 or more drugs	90 (90.0%)	154 (90.6%)	1.00
Proportion (%) of patients administering 10 or more drugs	39 (39.0%)	85 (50.0%)	0.10

^a^ median [minimum–maximum]. AC, active care; ECOG, Eastern Cooperative Oncology Group; NPO, nil per oral; UC, usual care.

**Table 4 jcm-12-07431-t004:** Major causative drugs and types of medication-related problems determined by network analysis.

Causative Drug	Cause *	Problem *
Lipid lowering agents	C1.2. No indication for drug	P3.1. Unnecessary drug-treatment
Pain managing agents	C1.1. Inappropriate drug according to guidelines/formulary	P2.1. Adverse drug event (possibly) occurring
Pain managing agents	C1.1. Inappropriate drug according to guidelines/formulary	P1.2. Effect of drug treatment not optimal
Neuropsychiatric drugs	C1.3. Inappropriate combination of drugs, or drugs and herbal medications, or drugs and dietary supplements	P2.1. Adverse drug event (possibly) occurring
Pain managing agents	C1.5. No or incomplete drug treatment in spite of existing indication	P1.2. Effect of drug treatment not optimal

* Each code for cause and problem domains is the category from PCNE version 9.1.

**Table 5 jcm-12-07431-t005:** Deprescribing rates of preventive medications.

Total Number of Prescriptions of Preventive Medications in the UC Group (N) (N = 100)	Deprescribing Rate of Preventive Medications in the UC Group (N, (%)) (N = 100)	Total Number of Prescriptions of Preventive Medications in the AC Group (N) (N = 170)	Deprescribing Rate of Preventive Medications in the AC Group (N, (%)) (N = 170)	*p*-Value
125	12 (10.4)	270	80 (29.6)	*<0.001*

**Table 6 jcm-12-07431-t006:** Clinical outcomes other than deprescribing rate of preventive medications.

Outcomes	UC Group (N = 100)	AC Group (N = 170)	Title 4
Proportion of patients deprescribed one or more MRPs at discharge (frequency, %)	3.00% (3/100)	40.0% (68/170)	<0.001 *
Changes of pill burden ^a^	0 [−9–10]	0 [−11, 7]	0.13
Change in proportion of patients with PP * at discharge (%)	−14%	−8.1%	0.87

^a^ median [minimum–maximum]. * Polypharmacy (PP) is defined as using 5 or more medications in total. AC, active care; MRP, medication-related problems; PP, polypharmacy; UC, usual care.

**Table 7 jcm-12-07431-t007:** Acceptance rate (%) of pharmacist’s intervention by I codes from PCNE version 9.1.

Intervention Domain in PCNE Classification	Usual Care (UC)		Active Care (AC)		*p*-Value *
	N (%)	Acceptance Rate (%)	N (%)	Acceptance Rate (%)	
**At prescriber level**	**16 (100)**	**68.8**	**199 (100)**	**78.9**	**0.021**
I1.1 Prescriber informed only	1 (6.2)	0	36 (18.0)	66.7	
I1.2 Prescriber asked for information	6 (37.5)	100	10 (5.0)	100	
I1.3 Intervention proposed to prescriber	9 (56.2)	55.6	143 (71.5)	81.8	
I1.4 Intervention discussed with prescriber	0 (0.0)	-	10 (5.0)	60	
**At patient level**	-	-	**23 (100)**	**100**	-
I2.1 Patient (drug) counseling	-	-	5 (2.5)	100	
I2.2 Written information provided (only)	-	-	-	-	
I2.3 Patient referred to prescriber	-	-	-	-	
I2.4 Spoken to family member/caregiver	-	-	18 (9)	100	
**At drug level**	**10 (100)**	**30.0**	**186 (100)**	**78.0**	**0.003**
I3.1 Drug changed to…	1 (6.2)	100	31 (15.5)	74.2	
I3.2 Dosage changed to…	2 (12.5)	0	21 (10.5)	85.7	
I3.3 Formulation changed to…	0 (0.0)	-	2 (1.0)	100	
I3.4 Instructions for use changed to…	0 (0.0)	-	5 (2.5)	100	
I3.5 Drug paused or stopped	6 (37.5)	66.7	116 (58.0)	75	
I3.6 Drug started	1 (6.2)	0	11 (5.5)	90.9	

* Each code for cause and problem domains is the category from PCNE version 9.1, and the total number of accepted interventions at each level of pharmacist’s interventions were compared between groups by Fisher’s exact test. PCNE, Pharmaceutical Care Network Europe.

## Data Availability

The dataset generated during and/or analyzed during the current study are available from the corresponding author on reasonable request. Requests to access the dataset should be directed to J.M.O., jmoh@snu.ac.kr.

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
