# Peer review of "Consultation-Based Deprescribing Service to Optimize Palliative Care for Terminal Cancer Patients"

_jcm, 2023, doi:10.3390/jcm12237431_

Round 1

Reviewer 1 Report

Comments and Suggestions for Authors

A very interesting paper highlighting the authors' experience with Pharmacist-led deprescribing service developed within their Consultation-Based Palliative Care Team (CB-PCT).

Can the authors elaborate on the make-up of this Pharmacist-led deprescribing service, their specific roles and work process? For example the number of Pharmacists in each team, the number teams, experience (years of service) of the Pharmacist, timing of patient consultation, timing of medication reviews and the work process (what does the Pharmacist do when he/she identifies an inappropriate or redundant prescription? Refer and consult the attending physicians?)

According to the authors, the findings of the Pharmacist-led deprescribing service in CB-PCT were found to be clinically significant and practically implementable, has such services been implemented in other specialties in the authors' institution?

Author Response

Response on the first comment: The CB-PCT was comprised of five members, including two physicians, one nurse, one social worker, and one pharmacist. The pharmacist was required to have at least three years of experience in oncology pharmaceutical care. We have elaborated on the details of the service process in the Materials and Methods section, specifically in subsection 2.1. Pharmacist-led deprescribing service of CB-PCT section as follows:

  • The team was comprised of two physicians, one nurse, one social worker, and one pharmacist, with the pharmacist having over three years of experience in oncology pharmaceutical care. The target populations for this pharmaceutical service included patients enrolled in the consultation-based palliative care team, along with healthcare providers from the wards caring for these patients, and the members of the palliative care team itself. Furthermore, there were four objectives for this deprescribing pharmaceutical service. First, it aimed to provide medication reconciliation to accurately ascertain the patient's medication history and minimize unintentional discrepancies during transition of care. Second, by offering comprehensive medication evaluation and deprescribing services, the goal was to identify and resolve medication-related problems in terminally ill cancer patients and to deprescribe medications with a low risk-benefit ratio. Third, it involved providing evidence-based drug information services to deliver accurate medical information to healthcare professionals. Finally, its objective was to ensure the continuity of pharmacist interventions through the application of discharge pharmaceutical care services.

We also added details regarding the time spent on patient consultations and the timing of the medication evaluations on page 3 to 4, from lines 141 to 152.

  • 2. Comprehensive Medication Evaluation and Deprescribing (CME&D): Initiated within 48 hours post CB-PCT enrollment, this core sub-service continued throughout the patients’ stay. CME&D offered deprescribing interventions and an in-depth medication assessment to identify MRPs, which include both evident problems and potentially inappropriate prescriptions. CME&D involved stages of finding, assessing, discussing, recommending, monitoring, and documenting. During the finding stage, patients’ medications were evaluated, and pharmacy consultations were provided to patients as needed, with pharmacists typically spending about 15 minutes with each patient. In the assessing stage, opportunities for deprescribing were identified primarily based on the previously developed deprescribing guidelines (SNUH deprescribing guidelines, Supplementary methods S2) and a variety of established guidelines, including Beers’ criteria [33] and STOPPFrail [22]. Detected MRPs were shared within the team and actively discussed, not only during CB-PCT weekly meetings, but also via the team’s instant messaging apps, ensuring timely interventions. Pharmacists intervened via short message service to physicians or through verbal communication. Following evaluation, the findings was documented in the electronic health records (EHR) using department-specific forms. Pharmacists evaluated patients’ prescriptions within 24 hours of notification by the team’s nurse of the patient’s enrollment. Aligned with the patient’s consultation cycle of the team, MRPs were tracked and monitored for resolution. Finally, the results of the interventions were recorded in the electronic health record and pharmacist’s database.

Response on the second comment: We added the following in the discussion section to further discuss another implemented services in our institution.

  • The consultation-based palliative care model is recognized for its effectiveness in delivering palliative care with limited resources. This approach has been a foundational aspect of many early palliative care models in the United States. At our institution, nutritional management serves as another exemplary implementation of this consultative model [43]. This indicates the versatility of the consultation-based model in addressing diverse patient care needs, demonstrating its broad applicability across different aspects of patient care.

Reviewer 2 Report

Comments and Suggestions for Authors

Thank you for giving me the opportunity to review this interesting manuscript on Consultation-based deprescribing service to optimize palliative care for terminal cancer patients. The study focuses on the relevant study results and the paper is easy to follow and does have a logical flow despite the very complex study design. It is well-written and focuses on an interesting topic that is clinically relevant. The title and abstract cover the main aspects of the work.

Nevertheless, I would like to address some concerns to help to improve the quality of the manuscript.

-       I would recommend checking all the abbreviations again. Some might not be necessary (such as MCMC) as you do not use several very often in the manuscript and in my opinion, it is easier to follow the text with fewer abbreviations; additionally, others were not written out before their first use (e.g. MRPs in the abstract or EMR).

-       I wondered what you mean by “expired” in line 243.

-       In Figure 3, I did not understand what the light grey and the darker grey bars show, please clarify.

-       Why did you choose a retrospective design instead of a prospective design?

Overall, this study makes a worthwhile contribution to the literature.

Author Response

Thank you for your insightful comments. Below are the revisions made in response to each of your points:

-  I would recommend checking all the abbreviations again. Some might not be necessary (such as MCMC) as you do not use several very often in the manuscript and in my opinion, it is easier to follow the text with fewer abbreviations; additionally, others were not written out before their first use (e.g. MRPs in the abstract or EMR).

  • Thank you for your valuable feedback regarding the use of abbreviations in our manuscript. Upon your recommendation, we have carefully reviewed all abbreviations. We agree that minimizing the use of less frequently mentioned abbreviations, such as MCMC, does indeed improve the readability of the text. Therefore, we have revised the manuscript to spell out terms that are not frequently used.

-   I wondered what you mean by “expired” in line 243.

  • Thank you for your inquiry regarding the use of the term ‘expired’ in our manuscript. In this context, ‘expired’ is used to indicate that patients passed away. According to the Merriam-Webster Dictionary, ‘expire’ means ‘to breathe one’s last breath’ and is used as an intransitive verb.

-  In Figure 3, I did not understand what the light grey and the darker grey bars show, please clarify.

  • Thank you for your comment regarding Figure 3, titled "Proportion (%) of Medication-Related Problems (MRPs) Identified by Drug Class in Each Group." In this figure, the light grey bars represented the proportion of MRPs identified in the usual care (UC) group, while the darker grey bars showed the proportion of MRPs found in the active care (AC) group. To improve readability and assist in better distinguishing between the two groups, we have increased the size of the legend and modified the color scheme. These adjustments will facilitate a clearer interpretation of the data presented in Figure 3, especially in differentiating the MRPs identified in each group. 

-   Why did you choose a retrospective design instead of a prospective design?

We chose this approach for several reasons. Firstly, starting in March 2020, every patient who was enrolled in the CB-PCT service received the newly developed pharmacist-led deprescribing service. For ethical reasons, it was not appropriate to withhold this already implemented service form certain patients for the sake of a prospective study design. Selectively providing the service to some patients while denying it to others could raise ethical concerns, especially since the service was already established and in practice. Secondly, to effectively compare the outcomes of this new service, we deemed a retrospective comparison with a historical control group, in this case, the usual care (UC) group, to be the most appropriate design. 

Reviewer 3 Report

Comments and Suggestions for Authors

Vitamins, lipid lowering agents, etc., should absolutely be targets for deprescribing, however "gastroprotective" agents, particularly PPIs are indicated to prevent GI bleeding in patients at high risk, and are used for symptom relief in patients with pyrosis and other conditions that arise from lower stomach pH.   How did the authors (and the pharmacists) distinguish between appropriate use of PPIs and appropriate deprescribing of these medications?

Appropriate use of PPIs includes prescribing for:

  • GERD 
  • Zollinger-Ellison Syndrome 
  • Peptic ulcer disease  
  • NSAID-associated ulcers 
  • H. pylori infection 
  • Eosinophilic esophagitis  
  • Acute upper GI bleeding
  •  

Author Response

Thank you for your insightful comments. As you rightly pointed out, gastroprotective agents, such as proton pump inhibitors (PPIs), are used for conditions like NSAID-associated ulcers or H. pylori infections. Thus, the deprescribing rate of preventive medications was calculated based only on those prescriptions identified for preventive use following the pharmacist’s evaluation on their indications. In our study, pharmacists rigorously evaluated the appropriateness of these prescriptions, particularly focusing on deprescribing, through medication reconciliation (MR) and comprehensive medication evaluation & deprescribing (CME&D) sub-service. In the supplementary data, I have included the deprescribing guidelines we previously developed for terminally ill patients with advanced cancer. Upon analyzing the indications for gastroprotective agents, we found that 80% of the medication-related problems (MRPs) induced by these agents were due to either no indication or unnecessary drug-treatment. Additionally, gastroprotective agents were identified as one of the drug classes with futility in a previous study conducted with Korean terminally ill cancer patients (as cited in [35]).

I have added the details in the method section regarding evaluation on the deprescribing rate of preventive medications as follows:

  • The deprescribing rate of preventive medications was calculated based only on those prescriptions identified for preventive use following the pharmacist’s evaluation of their indications.

I have also added the following information in the discussion section (from the line 422 to 424) to improve the conclusions supported by the results.

  • Specifically, 80% of the MRPs associated with gastroprotective agents were due to either lack of indication or unnecessary drug treatment.
